# Initial Evaluation of Uroplakins UPIIIa and UPII in Selected Benign Urological Diseases

**DOI:** 10.3390/biom11121816

**Published:** 2021-12-02

**Authors:** Beata Szymańska, Michał Matuszewski, Janusz Dembowski, Agnieszka Piwowar

**Affiliations:** 1Department of Toxicology, Faculty of Pharmacy, Wroclaw Medical University, 50-556 Wroclaw, Poland; agnieszka.piwowar@umw.edu.pl; 2Department of Urology and Oncological Urology, Faculty of Medicine, Wroclaw Medical University, 50-556 Wroclaw, Poland; michal.andrze.matuszewski@gmail.com (M.M.); janusz.dembowski@umw.edu.pl (J.D.)

**Keywords:** uroplakin UPIIIa, uroplakin UPII, benign urological diseases

## Abstract

Background: Uroplakins (UPs) are glycoproteins that play a specific role in the structure and function of the urothelium. Disorders which affect the normal expression of UPs are associated with the pathogenesis of infections and neoplasms of the urinary tract, primary vesicoureteral reflux, hydronephrosis and renal dysfunction. The appearance of uroplakins in the urine and/or plasma may be of potential importance in the detection of urinary tract dysfunction. The aim of the present study was to investigate uroplakin IIIa (UPIIIa) and uroplakin II (UPII) expression in patients with selected urological diseases. Methods: Plasma and urine from patients with benign prostatic hyperplasia (BPH), urethral stricture (US), urinary tract infection (UTI) and urolithiasis were compared to healthy people without urological disorders. UPs concentrations were measured by the immunoenzymatic method. Results: In patients with BPH and UTI, concentrations of UPIIIa in urine and plasma, as well as UPII in urine, were statistically significantly higher than in the control groups. In the US group, only the plasma UPIIIa concentration differed significantly from the control. Conclusion: The conducted research shows that benign urological diseases may affect the state of the urothelium, as manifested by increased concentrations of both UPs in patients’ urine and plasma, especially in BPH and UTI.

## 1. Introduction

The most superficial, single layer in direct contact with urine is made up of large umbrella cells. The main components of these umbrella cells are differentiated UPs [1,2]. In humans, four types of UPs are known: UPIa, UPIb, UPII, UPIII. The feature most often taken into account when classifying UPs is the number of transmembrane domains in the molecule. UPIa and UPIb have four transmembrane domains and belong to the tetraspanin family. In contrast, UPII and UPIII are simple, integral membrane proteins with a single transmembrane domain. UPs Ia, Ib and III are glycoproteins, and their sugar residues are very important elements in the synthesis process and are essential for their proper functioning. The mature form of UPII is devoid of sugar. UPII and UPIII have one transmembrane domain with a C-terminus in the plasma of the cells and an N-terminus in the intravesical space. Several isoforms of UPIII exist. The dominant isoform in urothelial plates is UPIIIa [3,4,5]. Research has shown that urothelial umbrella cells are heterogeneous, as some normal-looking umbrella cells can possess only one, instead of two, UP pairs. This heterogeneity is more visible in the urothelium of human ureter than that of the bladder, which proves that ureter urothelium is intrinsically different from bladder urothelium. However, further research is needed to clarify the physiological and pathological significance of the observed urothelial heterogeneity [6].

As much as 90% of the urothelium is covered with UPs; the remaining 10% is the exposed surface of the cell membrane that forms flexible areas. These regions, together with elements of the cytoskeleton, for example, are involved in folding and changing the capacity of the bladder [7,8]. The most obvious function of UPs is to create an impermeable blood-urine barrier and to protect deeper cells and other tissues from harmful substances (including ammonia, urea, modified drugs and toxins) in the urine collected by the bladder [9,10,11].

Disorders which affect the normal expression of UPs are strongly associated with the pathogenesis in infection and urinary tract malignancies, primary vesico-urinary reflux, hydronephrosis and renal impairment [10]. For this reason, UPs may become a clinically significant diagnostic or therapeutic target for many urological disorders [12,13,14].

Due to their selective tissue specificity, UPs are an attractive model in the search for tumor markers of the urothelium. The detection of their expression in the analyzed biological material indicates a high probability of a neoplasm of urothelial origin, which is important in the differentiation of anaplastic pelvic neoplasms [15,16,17]. Already in pre-neoplastic lesions, UP expression is reduced, although the lamellar structure remains preserved [18]. In noninfiltrating bladder tumors, accounting for approximately 50% of all bladder cancer (BC) cases, tissue fragments from neoplastic sites showed abnormal UP expression compared to normal tissues, indicating abnormal urothelial cell differentiation [19,20,21].

In the available literature, there are reports on UPs in the context of their participation in urinary tract infection (UTI) [22]. The most common etiology of UTI are Gram-negative Enterobacteriaceae, predominantly Escherichia coli, especially uropathogenic *E. coli* (UPEC) strains. Uropathogenic bacteria colonize the urinary tract thanks to interactions with species-specific lectins and fimbriae types 1 and P, as well as mannose-rich urothelial glycoconjugates. The expression of mannose-rich fragments is a more important factor in infection occurrence than the pathogenicity of *E. coli* strains themselves. Bacterial adhesion to the urothelium is possible through the interaction between adhesion molecules of FimH and mannose fragments of subunit UPIa of urothelial plaques [23]. Except for adhesion, the connection of UPEC fimbriae with UPIa/Ib starts reactions which allow *E. coli* to invade and create an intercellular population of bacteria. This population is able to proliferate and form intercellular bacterial communities, which are probably responsible for recurrent urinary tract infections because of their antibiotic resistance [24,25].

In recent years, new knowledge about the structure and functions as well as molecular pathways in the epithelium of the urinary tract has opened up new possibilities for noninvasive diagnostics and the development of therapies for targeted urinary tract diseases. At present, research is mainly focused on gaining knowledge of new, and improving our understanding of already known, tissue-specific biomarkers, which include UPs. As shown in the available literature, changes in UPs expression are associated with the pathogenesis of infection [22,23] or bladder cancer [15,16,17,18,19]. This inspired us to study these glycoproteins in other urological diseases where no similar studies have been conducted. Therefore, in our own research, UPIIIa and UPII measurements were carried out in the urine and plasma of patients with selected benign urological diseases.

One of the most common urological diseases in men is benign prostatic hyperplasia (BPH). The pathogenesis and progression of benign prostatic hyperplasia may be influenced by factors such as aging, hormonal changes, metabolic syndrome and inflammation. Although the prostate is a separate gland not related directly to urothelium, its location may cause destabilization of the urinary tract epithelium at this site and alter the expression of UPs [26].

Urolithiasis is a condition characterized by the formation of deposits from components in the urine. The most important causes of urolithiasis include genetic inheritance, nutrition, metabolic disorders (e.g., hypercalciuria and hyperoxaluria), environmental factors, and drugs that can induce the appearance of stones. Crystals formed in the urine attach to the urothelium, which may affect the subsequent growth of the stone [27].

Urethral stricture (US) is a condition of abnormal narrowing involving the part of the urethra that is surrounded by a spongy body. US is the result of ischemic fibrosis, which manifests as a scar on the spongy body. Stenosis can occur at any age, in both men and women. US has many different etiologies, but can be divided into four main categories: iatrogenic, idiopathic, trauma, and inflammation [28].

The aim of this study was to investigate changes in the concentration of UPIIIa and UPII in the urine and plasma of patients with selected benign diseases of the urinary tract, such as BPH, US, UTI and urolithiasis, in comparison to healthy people with excluded urological diseases, who constituted the control groups. The interrelationships between UPIIIa and UPII in the urine and plasma of patients were also investigated. The value of the tested UPs as potential diagnostic parameters in the presented urological diseases was also assessed.

## 2. Material and Methods

### 2.1. Patients and Control Groups

The study group consisted of patients with selected urinary diseases, such as BPH, US, UTI caused by UPEC and urolithiasis treated at the Urology and Oncological Urology Clinic (Wroclaw Medical University) during the period from 2015 to 2016. The control groups (C1 and C2) were selected from participants with no history of urological disorders, excluded by clinical examination of the cytology of urine sediment and a urine strip test. Group C1 consisted only of men, and group C2 consisted of both women and men. All participants were informed of the aim of the study and gave written consent to participate. The study was approved by the Ethics Committee of Wroclaw Medical University (KB-292/2-16).

### 2.2. Material

The material for laboratory test examination of UPIIIa and UPII consisted of plasma and urine from the patient groups and the control groups. Blood samples were taken into plastic tubes (BD Vacutainer, Franklin Lakes, NJ, USA) with anticoagulant agent (3.2% buffered sodium citrate). Morning urine samples were collected in polystyrene containers (Aptaca, Canelli, Italy). Tubes with blood and urine were centrifuged by an MPW-350 laboratory centrifuge (MPW Instruments, Warsaw, Poland) at 1500 rpm for 10 min, after which the obtained supernatant (plasma and urine) was removed, placed in Eppendorf tubes and stored at −80 °C for further investigation.

### 2.3. Method

UPIIIa and UPII concentrations were measured in plasma and urine by Enzyme-Linked Immunosorbent Assay (ELISA) using USCN Life Science Inc., (Cloud–Clone Corp.), Wuhan, China kits. All assays were performed strictly according to the manufacturer’s instruction. The microplate was precoated with an antibody specific to UPIIIa/UPII. Standards or samples (100 μL) were added to the appropriate microplate wells with a biotin-conjugated antibody specific to UPIIIa/UPII. Next, avidin conjugated to Horseradish Peroxidase (HRP) was added to each microplate well and incubated. Next TMB (3,3′,5,5′-tetramethylbenzidine) substrate solution was added, which caused only the wells that contained UPIIIa/UPII, a biotin-conjugated antibody, and the enzyme-conjugated avidin to display a change in color. The enzyme-substrate reaction was terminated by the addition of a stop solution, and the color change was measured spectrophotometrically at a wavelength of 450 nm by a Synergy HTX Multi-Mode Microplate Reader (BioTek Instruments, Winuski, VT, USA). The concentration of UPIIIa/UPII in the samples was then determined by reading the absorption of the samples to the standard curve.

Concentrations of UPIIIa and UPII in urine were calculated in relation to the urine creatinine concentration estimated by Jaffe’s routine method based on the reaction of picric acid (Picric Acid, Saint Louis, MO, USA, SIGMA, Cat. No. 319287). Results were reported in ng/mL for UPs in plasma and ng/mg creatinine for UPs in urine. The determined values of the concentration of UPs in the urine were converted to the concentration of urinary creatinine in order to eliminate the influence of dilution or concentration.

### 2.4. Statistical Analysis

Statistical analyses were conducted with Statistica PL software (version 13.3). The normality of distribution was checked by Lilliefors and Kolmogorov-Smirnov tests. Student’s t and one-way ANOVA analysis of variance with Tukey’s post-hoc test for parametric data were used for the appropriate variables. The associations between UPs were analyzed by the Pearson test. The receiver operating characteristic curve (ROC) was estimated. The area under the curve (AUC) and cut-off point were calculated. Diagnostic value indicators with 95% CI, such as sensitivity and specificity, were calculated. In all analyses *p* < 0.05 was accepted as a significant value.

## 3. Results

### 3.1. Patient and Control Characteristics

The study groups consisted of patients with selected urological diseases of the urinary tract such as BPH, US, UTI and urolithiasis, and included men and women. The groups of patients with BPH and US were male only, while the groups of patients with UTI and urolithiasis were formed of both women and men. Therefore, two control groups, i.e., C1 (consisting only of men) and C2 (consisting of both men and women), were created. No statistically significant differences in characteristic features were observed between the patient and control groups. The age and sex of patients with the selected urinary diseases and the control group were not different (*p* > 0.05). The demographic and clinical characteristics of the examined groups are shown in Table 1.

Analyses of the results were carried out in two parts. The first analysis of the UP results concerned patients with BPH, US and the C1 control group, composed of men. The second analysis of the UP results concerned patients with UTI, urolithiasis and C2, which were both male and female.

### 3.2. UPIIIa and UPII in the Urine and Plasma of Patients with BPH, US and Control Group C1

The mean concentrations and standard deviation of UPIIIa and UPII in the urine and plasma of patients with BHP, US and control group (C1) are presented with statistical analysis in Table 2.

Statistically significant differences were found in our analyses of groups of patients with BPH and US, as well as in the C1 group, for UPIIIa and UPII in both urine and plasma (one-way ANOVA test), (Table 2).

The mean urinary and plasma concentrations of UPIIIa in BPH patients were 2.3-fold and 3.5-fold higher than in the control group (C1), respectively. UPIIIa in both urine and plasma of BPH patients showed a statistically significant difference compared to the control group. The mean urinary and plasma concentrations of UPII in BPH patients were 4.3-fold and 1.2-fold higher than in the control group, respectively. UPII in the urine of BPH patients was statistically significantly higher than in the control group, but in plasma, UPII did not differ significantly relative to the C1.

The mean urinary and plasma concentrations of UPIIIa in US patients were 1.4-fold and 3.2-fold higher than in the control group (C1), respectively. In patients with US, mean values of plasma UPIIIa were higher compared to the concentrations of UPIIIa obtained in the group C1, but there was no significant difference between the concentrations of UPIIIa and C1 in urine. The mean urinary and plasma concentrations of UPII in US patients were 2.9-fold and 1.4-fold higher than in the control group, respectively. There was no significant difference between the values of UPII concentration in patients with US compared to C1 in urine and plasma.

### 3.3. UPIIIa and UPII in the Urine and Plasma of Patients with UTI, Urolithiasis and Control Group C2

The mean concentrations and standard deviation of UPIIIa and UPII in the urine and plasma of patients with UTI, urolithiasis and control group (C2) are presented with statistical analysis in Table 3.

Statistically significant differences were found in our analyses of groups of patients with UTI and urolithiasis, as well as in the C2 group, for UPIIIa in urine and plasma and UPII in urine (Table 3).

The mean urinary and plasma concentrations of UPIIIa in UTI patients were 1.6-fold and 3.1-fold higher than in the control group, respectively. UPIIIa in both urine and plasma of UTI patients showed a statistically significant difference compared to the control group (C2). The mean urinary and plasma concentrations of UPII in UTI patients were 3.2-fold and 1.2-fold higher than in the control group (C2), respectively. UPII in the urine of UTI patients was statistically significantly higher than in the control group, but the plasma UPII did not differ significantly from the value obtained in C2.

The mean urinary and plasma concentrations of UPIIIa in urolithiasis patients were 1.5-fold and 2.9-fold higher than in the control group, respectively. The mean urinary and plasma concentrations of UPII in urolithiasis patients were 2.4-fold and 1.2-fold higher than in the control group (C2), respectively. In the group of patients with urolithiasis, both UPIIIa and UPII in urine and plasma showed no statistically significant differences compared to the control group C2.

Visualizations of the obtained results of UPIIIa and UPII in urine and plasma of patients with selected benign urological diseases are presented in Figure 1.

There were no statistically significant differences between the values of UPIIIa and UPII concentrations in the urine and plasma of patients with selected benign urological diseases in relation to each other.

### 3.4. Mutual Correlations between UPIIIa and UPII

Mutual positive correlations between UPIIIa (urine) and UPIIIa (plasma) in BPH and UTI were demonstrated. Such correlations were not found in the case of UPII. Urine UPIIIa correlated with urine UPII in patient groups with BPH and UTI (Table 4).

### 3.5. ROC Curves Analysis for UPIIIa and UPII in Selected Benign Urological Diseases

The diagnostic value of the examined UPIIIa and UPII in both biological fluids (plasma and urine) was evaluated. Figure 2 shows the receiver operating curves (ROC) for UPIIIa and UPII, which obtained the best diagnostic values. A ROC curve analysis of UPIIIa (BPH-plasma), UPIIIa (UTI-plasma), UPIIIa (BPH-urine), UPIIIa (UTI-urine), UPII (BPH-urine) and UPII (UTI-urine) conducted for examined UPs showed an area under the curve (AUC) of 99% [95% confidence interval (CI) (99–100%), *p* < 0.001), sensitivity 100%, specificity 99%], 99% [95% CI (99–100%, *p* < 0.001), sensitivity 99%, specificity 97%], 89% [95% CI (79–99%, *p* < 0.001), sensitivity 92%, specificity 88%], 79% [95% CI (65–93%, *p* < 0.001), sensitivity 82%, specificity 75%], 74% [95% CI (58-89% *p* = 0.003), sensitivity 99%, specificity 72%], 63% [95% CI (45-79% *p* = 0.162), sensitivity 84%, specificity 51%], respectively (Figure 2).

The cut-off values for UPIIIa (BPH-plasma), UPIIIa (UTI-plasma), UPIIIa (BPH-urine), UPIIIa (UTI-urine), UPII (BPH-urine) and UPII (UTI-urine) were 1.46 ng/mL, 1.29 ng/mL, 1.26 ng/mg creatinine, 1.17 ng/mg creatinine, 0.13 ng/mg creatinine, 0.11 ng/mg creatinine, respectively (Figure 2).

## 4. Discussion

In the presented study, an attempt was made to evaluate the changes in UPIIIa and UPII expression in selected urological benign diseases, i.e., BPH, US, UTI and urolithiasis, by measuring the concentration of these proteins in the urine and plasma of patients.

The results were related to the control group, which consisted of people of a similar age to the patients without urological diseases.

A statistically significantly higher concentration of UPIIIa was found in the urine and plasma of the group of patients with BPH and UTI compared to the concentration of this UPs in the control groups (*p* < 0.001). The highest concentrations of UPIIIa were obtained in the urine of patients (2.22 ng/mg creatinine, *p* < 0.001) and plasma (2.07 ng/mL, *p* < 0.001) with BPH.

The lowest concentration of UPIIIa was found and in the urine of US patients (1.39 ng/mg creatinine) and in the plasma of patients with urolithiasisis (1.78 ng/mL).

The concentrations of urinary UPII in patients with BPH and UTI were statistically significantly higher compared to the values of this uroplakin in the control (*p* < 0.001).

Our study revealed significant correlations between plasma UPIIIa and urinary UPIIIa concentrations in patients with BPH and UTI, and between urinary UPIIIa and UPII in BPH and UTI. This may indicate damage caused by BPH to the structure of the urothelium, manifested by a significant increase in UPIIIa in both the plasma and urine of patients.

UPIIIa determined in patients’ plasma was characterized by a high diagnostic value, especially in BPH and UTI patients (based on high AUC). No statistically significant difference was found between the concentrations of UPIIIa in the groups of patients with BPH and UTI, which precludes the possibility of distinguishing these diseases. However, the potential value of UPIIIa as a diagnostic indicator in the presented urological diseases should be confirmed by performing measurements in groups with a larger number of patients.

In the available literature, the most attention is paid to UPs in BC. UPIIIa is a moderately sensitive but specific marker in the immunohistochemical analysis of primary and metastatic BC. It is also a proposed marker for the differentiation of primary BC from other primary cancers of the genitourinary system [29]. UPII as a marker is characterized by exceptional specificity in detecting BC in the blood, and can be used to assess the stage of the neoplastic process or its metastasis, as well as to monitor treatment [30].

Tsumura et al. [31] measured the serum concentration of UPIII in patients with BC, they observed a significant increase in serum UPIII levels in patients compared to healthy controls.

Lai et al. [32] showed an increased amount of UPIIIa in the urine of patients with mild urothelial lesions compared to healthy subjects, but with a lower concentration of UPIIIa compared to patients with BC. The results obtained by the authors were expressed only by the A/µg (absorbance index of total protein in urine), without specifying the concentration converted to creatinine.

Our earlier studies of UPs in patients with BC showed a significant increase in the concentrations of UPIIIa and UPII in both urine and plasma. Previous studies on UPIIIa in patients with BC showed significantly higher levels of UPIIIa in the plasma and urine of patients with bladder cancer compared to controls, but the malignancy and invasiveness of the cancer made no significant difference. The concentration of UPIIIa in the urine of patients with BC was higher than the concentration of UPIIIa in currently analyzed, non-neoplastic urological diseases [33]. The concentrations of UPII in patients with BC were statistically significantly higher compared to the control, and amounted to 3.87 ng/mL in plasma (*p* < 0.001) and 0.50 ng/mg creatinine in urine [34]. Our study showed that the average concentration of UPII in selected urological diseases was lower than the values obtained in UPII in bladder cancer. No information has been found in the literature on the determination of UPII in other urological diseases.

The composition of the lipids associated with the urothelial plaques is also believed to play a large role in determining the permeability barrier of the uroepithelium. Urothelial plaques work in conjunction with specialized lipid microdomains (lipid rafts). Lipid rafts are regions of the plasma membranes enriched in cholesterol, sphingolipids and glycolipids, and may contain a protein, caveolin-1. These microdomains possess a versatile endocytic capacity that was recently implicated in microbial pathogenesis [35]. Duncan et al. [36] wanted to explain the molecular basis for *E. coli* invasion of the bladder epithelium by employing human bladder epithelial cells. They made the following observations: intracellular *E. coli* was associated with caveolae and lipid raft components; RNA reduction of caveolin-1 expression inhibited bacterial invasion; a signaling molecule required for *E. coli* invasion was located in lipid rafts and physically associated with caveolin-1; and bacterial invasion was inhibited by lipid raft disrupting/usurping agents. In the mouse bladder, the *E. coli* type 1 fimbrial receptor, uroplakin Ia, was located in lipid rafts, and lipid raft disruptors inhibited E. coli invasion. Cumulatively, *E. coli* uroepithelial invasion occurs through lipid rafts, which, paradoxically, contribute to bladder impermeability.

Thumbicat et al. [37] reported a signaling role for UPIIIa, the only major UP with a potential cytoplasmic signaling domain, in bacterial invasion and apoptosis. In response to FimH adhesin binding, the UPIIIa cytoplasmic tail undergoes phosphorylation on a specific threonine residue by casein kinase II, followed by an elevation of intracellular calcium. Pharmacological inhibition of these signaling events abrogates bacterial invasion and urothelial apoptosis in vitro and in vivo. The authors suggested that bacteria-induced UPIIIa signaling is a critical mediator of the pathogenic cascade induced in the host cell, and identified a novel therapeutic target for intervention in UTI pathogenesis.

Song et al. [38] reported a distinct TLR4-mediated mechanism in bladder epithelial cells (BECs). TLR4 is a transmembrane protein, a member of the toll-like receptor family, which belongs to the pattern recognition receptor family. Its activation leads to an intracellular signaling pathway NF-κB and inflammatory cytokine production which is responsible for activating the innate immune system. The authors showed that uropathogenic type 1 fimbriated Escherichia coli, and Klebsiella pneumoniae invaded BECs of TLR4 mutant mice in 10-fold or greater numbers compared to control. TLR4 mediated suppression of bacterial invasion was linked to increased intracellular cAMP levels which negatively impacted Rac-1 mediated mobilization of the cytoskeleton. Artificially increasing intracellular cAMP levels in BECs of TLR4 mutant mice restored resistance to type 1 fimbriated bacterial invasion. This finding revealed a novel function for TLR4, and another facet of bladder innate defense.

Our research shows that changes in UPIIIa concentrations in both plasma and urine are most noticeable in patients with BPH. The result is quite surprising, due to the fact that the prostate gland is not made of UPs. It is probably influenced by the location of the prostate, the enlargement of which causes compression of the prostatic segment of the urethra, and thus, may constitute a bladder obstruction (BPO-benign prostatic obstruction) and cause symptoms and ailments of the lower urinary tract such as bladder paralysis (LUTS-lower urinary tract symptoms) [39,40]. This can result in urothelial dysfunction, both in the urethra and in the bladder, leading to an increase in UPIIIa concentration, especially in the urine. It should also be remembered that lower urinary tract infections may occur more frequently, and are associated with difficulties in urinating properly [22].

Romih et al. [41] investigated the expression of UPs in the asymetric unit membrane (AUM) of the bladders of patients with obstruction of the outflow of the bladder. Bladder epithelial samples were examined by light and electron immunocytochemistry. Urothelial areas of the urinary tract without AUM were found in all bladder samples. UP-positive cells had the appearance of terminally differentiated umbrella cells, while cells from the UP-negative regions were undifferentiated. In addition, some urinary tract epithelial cells lacked AUM, which indicated a violation of the blood–urine permeability barrier. An increase in expression was observed in undifferentiated zones of the urinary tract epithelium, indicating that changes in the differentiation of the bladder urothelium may be due to obstruction of the urinary outflow.

Cho et al. [42] evaluated changes in the expression of UPs in the urothelium of patients with ulcerative interstitial cystitis/bladder pain syndrome (IC/BPS). Bladder samples were collected from patients with ulcerative IC/BPS who were treated with augmentation ileocystoplasty and from control patients. The expression levels of UPIb and UPIII in the urothelium were compared between the IC/BPS patients and control patients. Immunofluorescence staining showed that UPIb and UPIII were localized in the urothelium. Upon Western blot analysis, the expression of UPIII was found to be significantly increased in the IC/BPS group compared with the control group. However, expression of UPIb did not differ significantly between the IC/BPS and control groups.

Danka et al. [43] conducted an experiment involving the transurethral infection of wild-type mice and mice deficient in cathelicidin (antimicrobial peptide) with an uropathogenic strain of Escherichia coli. Cathelicidin-deficient mice showed no fundamental changes in immune activation, urinary tract epithelial structure, or intracellular bacterial growth rate. In addition, cathelicidin-deficient mice showed less intense cytokine responses, decreased neutrophil infiltration, and accelerated uroepithelial regeneration. The studies showed that UPIa and UPIIIa were equally expressed in bladder homogenates in both mouse strains. An immunohistological evaluation showed that uninfected bladders of wild-type and cathelicidin-deficient mice were similar in size, structure and tissue organization.

A study by Munnipali et al. [44] analyzed UPs in the male reproductive system of rats. MRNA of UPIa, UPIb, UPII and UPIIIb and their corresponding proteins were abundantly expressed in cabbage, tail, testes, seminal vesicles and prostate. UPs expression was also demonstrated in sperm. To investigate the role of UPs in innate immunity, UPs mRNA expression in response to endotoxin (lipopolysaccharide) challenge in vitro and in vivo was assessed. In rat testicular and epididymal lines, UPs mRNA levels increased in response to a lipopolysaccharide challenge. However, in the cabbage, tail, testes, seminal vesicle and prostate, UPs mRNA expression was significantly reduced. The results demonstrated a role for UPs in male reproductive physiology and innate immune responses. Among the studied urological diseases, only urolithiasis showed no significant differences in the concentration of UPs in urine and plasma compared to the control groups. There is no available literature on UPs conducted in patients with US or urolithiasis. Only one publication concerned animal studies of UPs with regard to urolithiasis.

Bilbaut et al. [45] used an animal model to study the early events of stone formation. Mice were supplemented with vitamin D and given water containing hydroxyl-L-proline, ammonium chloride and calcium chloride. Changes in the proliferation of urothelium cells and a reduction in UPIII expression in the study group were demonstrated. The authors suggested that urothelial proliferation may be a key trigger of stone formation.

The conducted research showed that the benign diseases presented in this study may affect the state of the urothelium, as manifested by an increased concentration of UPs in the urine and plasma of patients, especially in BPH and UTI. The increase in the concentration of UPs in urine and plasma is most likely related to the disease process in the urothelium. The presence of statistically significant differences between the values of UPIIIa in urine and plasma, or UPII in urine, in relation to the controls (based on high AUC), especially in BPH and UTI, indicates the potential diagnostic value of these proteins in detecting these diseases. However, the lack of statistically significant differences between the concentrations of UPs in selected urological diseases excludes UPs as markers differentiating these diseases. The obtained preliminary results of UPs in the urine and plasma of patients with selected urological diseases showed that these proteins may be helpful as potential biomarkers for monitoring the course of urological diseases and assessing the effectiveness of treatment thereof.

## 5. Conclusions

The present study showed that in the presented urological diseases, i.e., benign prostatic hyperplasia urethral stricture, urinary tract infection and urolithiasis, there are changes in the concentrations of UPIIIa and UPII, both in the plasma and urine of patients compared to healthy people, especially in patients with BPH and UTI. As a result of the processes taking place within the urinary tract, the structure of the urothelium is disturbed, as evidenced by the higher UPs concentrations. UPs could play a role as potential biomarkers for assessing the course of urological diseases and the effectiveness of their treatment. The innovative nature of the research and the limited knowledge on UPIIIa and UPII in urological, non-neoplastic diseases seem to warrant the continuation of this research, both for pre-investigated urological diseases and for other diseases of the urinary tract.

## Figures and Tables

**Figure 1 biomolecules-11-01816-f001:**
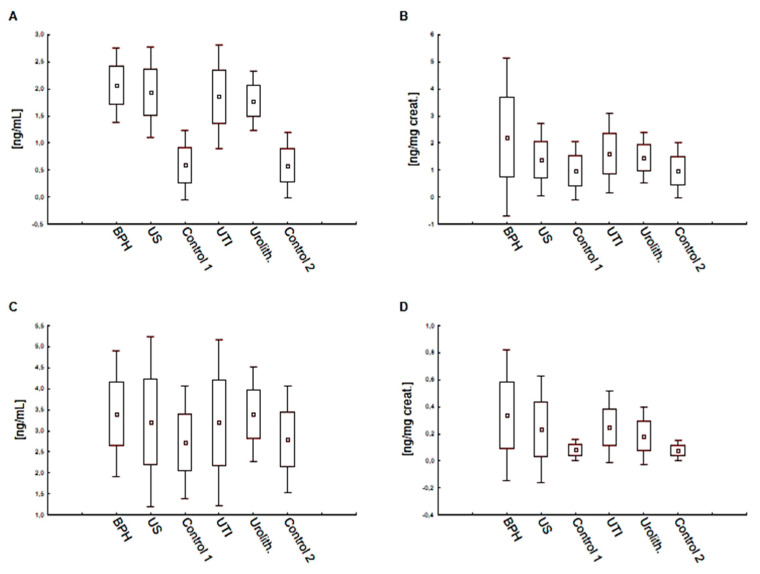
Mean values of concentrations, range of values and standard deviation of UPIIIa and UPII in the plasma and urine of patients with selected benign urological diseases and control groups C1 and C2: (**A**) UPIIIa (plasma); (**B**) UPIIIa (urine); (**C**) UPII (plasma); (**D**) UPII (urine).Abbreviations: BPH—benign prostatic hyperplasia; US—urethral stricture; UTI—urinary tract infection; Urolith.—Urolithiasis; C1—control group (men); C2—control group (men and women).

**Figure 2 biomolecules-11-01816-f002:**
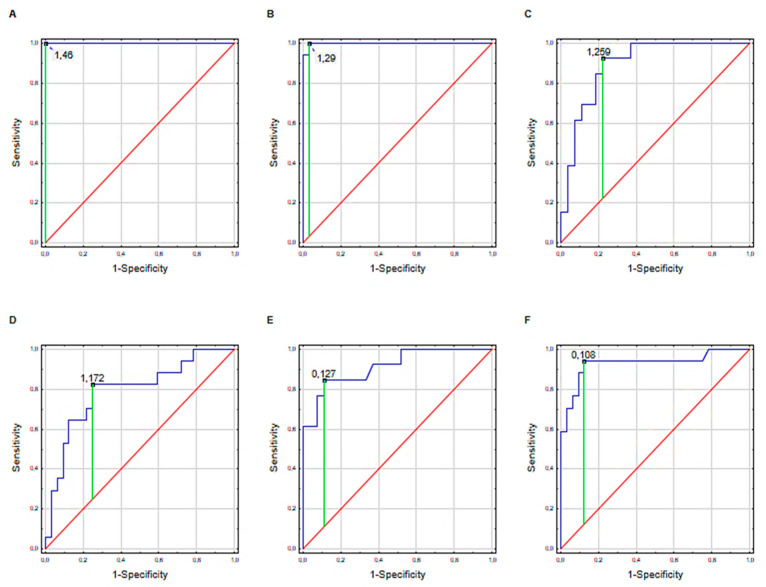
ROC curves analysis for UPIIIa and UPII in selected urological diseases: (**A**) UPIIIa (BPH-plasma), (**B**) UPIIIa (UTI-plasma), (**C**) UPIIIa (BPH-urine), (**D**) UPIIIa (UTI–urine), (**E**) UPII (BPH-urine), (**F**) UPII (UTI-urine). Abbreviations: BPH—benign prostatic hyperplasia; US—urethral stricture; UTI—urinary tract infection.

**Table 1 biomolecules-11-01816-t001:** Demographic and clinical data of patients and the control groups.

Population Characteristic	Patients	Control Groups
**N (%)**	44 (100)	32 (100); (C2)
Age, y, (range)	62 (21–83)	63 (40–78)
Man (%)	36 (82)	27 (84); (C1)
Age, y, (range)	63 (21–83)	64 (45–82)
Women (%)	8 (18)	5 (16)
Age, y, (range)	57 (28–73)	59 (42–71)
**BPH**		
Man (%)	13 (100)	
Age, y, (range)	69 (55–83)	
**US**		
Man (%)	7 (100)	
Age, y, (range)	53 (21–67)	
**UTI**		
N (%)	17 (100)	
Man (%)	12 (71)	
Age, y, (range)	62 (33–73)	
Women (%)	5 (29)	
Age, y, (range)	64 (50–73)	
**Urolithiasisis**		
N (%)	7(100)	
Man (%)	4 (57)	
Age, y, (range)	60 (48–67)	
Women (%)	3 (43)	
Age, y, (range)	47 (25–73)	

Abbreviations: BPH—benign prostatic hyperplasia; US—urethral stricture; UIT—urinary tract infection; C1— control group (man); C2—control group (man and woman); N—number of patients; y—years.

**Table 2 biomolecules-11-01816-t002:** Results for UPIIIa and UPII in patients with BPH, US and C1.

UPs	BPH Mean ± SD	US Mean ± SD	C1 Mean ± SD	*p**	Post-Hoc
UPIIIa (urine) [ng/mg cr.]	2.22 ± 1.48	1.39 ± 0.63	0.98 ± 0.56	<0.001	BPH:C1 < 0.001 US:C1 < 0.001 BPH:US = NS
UPIIIa (plasma) [ng/mL]	2.07 ± 0.35	1.94 ± 0.43	0.59 ± 0.33	<0.001	BPH:C1 < 0.001 US:C1 < 0.001 BPH:US = NS
UPII (urine) [ng/mg cr.]	0.34 ± 0.25	0.23 ± 0.20	0.08 ± 0.04	<0.001	BPH:C1 < 0.001 US:C1 = NS BPH:US = NS
UPII (plasma) [ng/mL]	3.41 ± 0.76	3.21 ± 1.73	2.79 ± 0.65	0.019	BPH:C1 = NS US:C1 = NS BPH:US = NS
urinary creatinine [mg/mL]	1.10 ± 0.48	0.82 ± 0.38	1.02 ± 0.25	NS	BPH:C1 = NS US:C1 = NS BPH:US = NS

*p**-values were calculated using one-way ANOVA analysis of variance; Abbreviations: BPH—benign prostatic hyperplasia; US—urethral stricture; SD—standard deviation; C1—control group (men).

**Table 3 biomolecules-11-01816-t003:** Results for UPIIIa and UPII in patients with UTI, urolithiasis and control group C2.

UPs	UTI Mean ± SD	Urolithiasisis Mean ± SD	C2 Mean ± SD	*p**	Post-Hoc
UPIIIa (urine) [ng/mg cr.]	1.62 ± 0.75	1.46 ± 0.47	0.99 ± 0.52	0.002	UTI:C2 < 0.001 Urol.:C2 = NS UTI:Urol. = NS
UPIIIa (plasma) [ng/mL]	1.86 ± 0.49	1.78 ± 0.28	0.59 ± 0.31	<0.001	UTI:C2 < 0.001 Urol.:C2 = NS UTI:Urol. = NS
UPII (urine) [ng/mg cr.]	0.25 ± 0.14	0.19 ± 0.11	0.08 ± 0.04	<0.001	UTI:C2 < 0.001 Urol.:C2 = NS UTI:Urol. = NS
UPII (plasma) [ng/mL]	3.19 ± 1.01	3.39 ± 0.57	2.72 ± 0.68	NS	UTI:C2 = 0.NS Urol.:C2 = NS UTI:Urol. = NS
urinary creatinine [mg/mL]	0.96 ± 0.52	0.93 ± 0.26	1.04 ± 0.23	NS	UTI:C2 = 0.NS Urol.:C2 = NS UTI:Urol. = NS

*p**-values were calculated using one-way ANOVA analysis of variance; Abbreviations: UIT–urinary tract infection; Urol.–Urolithiasis; SD–standard deviation; NS–not statistically; significant; C2–control group (men and women).

**Table 4 biomolecules-11-01816-t004:** Correlations between UPIIIa and UPII in patients with selected urological diseases.

Correlations between UPIIIa and UPII	r	*p**
BPH–UPIIIa (plasma) vs. UPIIIa (urine)	0.59	0.03
UTI–UPIIIa (plasma) vs. UPIIIa (urine)	0.61	0.01
BPH–UPIIIa (urine) vs. UPII (urine)	0.64	0.01
UTI–UPIIIa (urine) vs. UPII (urine)	0.50	0.04

*p**-significant difference (Pearson test); r-Pearson correlation coefficient; Abbreviations: BPH—benign prosaic hyperplasia; UTI—urinary tract infection.

## Data Availability

The datasets generated during and/or analyzed during the current study are available from the corresponding author on reasonable request.

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
