# Peer review of "Initial Evaluation of Uroplakins UPIIIa and UPII in Selected Benign Urological Diseases"

_biomolecules, 2021, doi:10.3390/biom11121816_

Round 1

Reviewer 1 Report

The paper is very well presented in form: introduction, methods, results.
But I did not understand the innovation and usefulness in the evaluation of UPIIIa and UPII in urological non-neoplastic diseases.

Author Response

Response to Reviewer 1 Comments

The paper is very well presented in form: introduction, methods, results.
But I did not understand the innovation and usefulness in the evaluation of UPIIIa and UPII in urological non-neoplastic diseases.

Response:

Thank you very much for this positive opinion. New changes to the manuscript have been marked up using the “Track Changes”. On behalf of the co-authors and myself, I will try to explain the Reviewer's doubts regarding the innovation and usefulness of our research.

Currently, diagnostics of urinary tract diseases, especially neoplastic diseases, consists in the use of endoscopic and imaging tests (ultrasound, computed tomography and others). In recent years, knowledge of the structure and function as well as molecular pathways conducted in the epithelium of the urinary tract open up new possibilities for non-invasive diagnostics and the possibility of defining therapies for targeted urinary tract diseases. Research tasks mainly concern the knowledge of new and improvement of already known tissue-specific biomarkers, which include uroplakins.

Most of the data available from the literature for detecting the expression of uroplakins for diagnostic purposes are immunocytochemical methods. Few studies on bladder cancer (BC) also presented quantitative analysis of UPs in the urine and plasma of patients using the enzyme immunoassay method.

We have not found similar studies on the level of UPs in mild urinary tract diseases in the available literature. We were interested in whether there will be differences in the level of UPs in patients in selected diseases such as benign prostatic hyperplasia (BPH), urethral stricture (US), urinary tract infection (UTI) and urolithiasis. We consider conducting such studies to be innovative, because they provided new information on the impact of diseases other than BC on the presence or absence of urothelial dysfunction, manifested by changes in the concentration of these proteins in urine or plasma.

Our research has shown that, especially in BPH and UTI, there is a significant increase in the concentration of UPs in both urine and plasma of patients. Therefore, in order to assess the usefulness of UPs as potential diagnostic biomarkers, the ROC analysis was performed. However, the lack of significant differences between the concentration of UPs in the presented diseases excludes these proteins as markers of detection and / or differentiation of these diseases, which is also a novelty in the topic of UPs.

Reviewer 2 Report

Well done study an uroplakins in urology. 

Major: please think about shortening introduction. It is somewhat lengthy and should be focused more on the study.

Minor: please check and correct table 1. Patients are present twice. The numbers are not always congruent. UTI Men n/% 4(9), Women n/%: 3/7? Is that correct? 

Author Response

Response to Reviewer 2 Comments

Well done study an uroplakins in urology. 

Major: please think about shortening introduction. It is somewhat lengthy and should be focused more on the study.

Minor: please check and correct table 1. Patients are present twice. The numbers are not always congruent. UTI Men n/% 4(9), Women n/%: 3/7? Is that correct? 

Response:

Thank you very much for this positive opinion. New changes to the manuscript have been marked up using the “Track Changes”.

Major: In line with the Reviewer's remark, the introduction has been shortened. The most important information on the topic of UPs in our research has been left. We also introduced changes to the numbering of the quoted references in the text and in the list of references at the end of the text.

Minor: Table 1, which was duplicated by mistake, has been corrected.

The percentage of women and men in the group of patients with urolithiasis was improved.

was: men 4 (9), women 3 (7);   (Table 1)

is: men 4 (57), women (43);  (Table 1)

Reviewer 3 Report

The authors conducted the present study to investigate changes in the concentration of UPIIIa and UPII in the urine and plasma of patients with selected benign diseases of the urinary tract, in comparison to healthy people as control. The interrelationships between UPIIIa and UPII in the urine and plasma of patients were also investigated. The value of the tested UPs as potential diagnostic parameters in the presented urological diseases was also assessed.

The authors need to address the following points:

#1: Line 46-47; “AUM (asymptomatic unit membrane)” should be written as “asymptomatic unit membrane (AUM)”.

#2: Line 106; “Ups” should be written as “UPs”.

#3: Line 127-128; This sentence should be grammatically revised.

#4: Materials and methods; I think that patients with BPH sometimes suffer from urinary tract infection. Did you recruit patients with BPH without UTI in this study?

#5: Materials and methods; The authors recruit patients with UTI caused by uropathogenic E. coli (UPEC). Patients with UTI were sometimes infected by other bacteria simultaneously. Did you recruit patients with UTI by UPEC only?

#6: Lines 164-165; Is ELISA an abbreviation of immunoenzymatic method? I think that Enzyme-Linked Immunosorbent Assay can be abbreviate to ELISA.

#7: Table 1; A lower part of Table 1 seems to a duplication of upper part of Table 1.

#8: Urolithiasis contains renal stone, ureter stone, bladder stone, and urethral stone. Did concentrations of UPs change according to the position of urolith?

#9: Table 2: At the first column, BHP should be revised as BPH.

#10: Table2 and Table 3: P values should be provided in comparison to control versus each pathological condition.

#11: Line261-263; In relation to the comment of #10, did this sentence correctly explain the contents of Table 3? Except for plasma UPII, all the p values were less than 0.05.

#12: Figure 1: I think that Figure 1 repeats the explanation of Table 2 and Table 3.

#13: After treatment of benign diseases of the urinary tract, will UPs decrease to normal range?

#14: How did you estimate sample size to investigate the utility of UPs in the condition of benign diseases of the urinary tract?

Author Response

Response to Reviewer 3 Comments

The authors conducted the present study to investigate changes in the concentration of UPIIIa and UPII in the urine and plasma of patients with selected benign diseases of the urinary tract, in comparison to healthy people as control. The interrelationships between UPIIIa and UPII in the urine and plasma of patients were also investigated. The value of the tested UPs as potential diagnostic parameters in the presented urological diseases was also assessed.

Response:

Thank you very much for this opinion. New changes to the manuscript have been marked up using the “Track Changes”.

The authors need to address the following points:

#1: Line 46-47; “AUM (asymptomatic unit membrane)” should be written as “asymptomatic unit membrane (AUM)”.

Response 1: This fragment of the article with the abbreviation AUM has been deleted, but of course we will remember how to save it correctly.

#2: Line 106; “Ups” should be written as “UPs”.

Response 2: The shortcut has been corrected.

#3: Line 127-128; This sentence should be grammatically revised.

Response 3: The incorrectly worded sentence: ,, Tsumura et al. [24] serum levels of UPIII in patients with urothelial carcinoma were measured. '' Has been replaced with: ,, Tsumura et al. [24] measured the serum concentration of UPIII in patients with urothelial bladder cancer showed a significant increase in serum uroplakin III levels in patients with BC compared to healthy controls.

#4: Materials and methods; I think that patients with BPH sometimes suffer from urinary tract infection. Did you recruit patients with BPH without UTI in this study?

Response 4: Yes of course. Patients with BPH, but also patients with US and urolithiasis were initially tested for urinary tract infections. Patients with the above-mentioned diseases with co-infection of the urinary system did not participate in the study.

#5: Materials and methods; The authors recruit patients with UTI caused by uropathogenic E. coli (UPEC). Patients with UTI were sometimes infected by other bacteria simultaneously. Did you recruit patients with UTI by UPEC only?

Response 5: Thank you very much to the reviewer for this comment. Yes of course. Based on the obtained microbiological results (cultures), we distinguished patients with UPEC infection. We were of particular interest to this group as literature data show the involvement of UPs in chronic E. coli infections. Of course, in our further research we will analyse patients with other urinary infections.

#6: Lines 164-165; Is ELISA an abbreviation of immunoenzymatic method? I think that Enzyme-Linked Immunosorbent Assay can be abbreviate to ELISA.

Response 6: The sentence  ,,Concentrations of UPIIIa and UPII were measured in plasma and urine by the immunoenzymatic method (ELISA) with Enzyme-Linked Immunosorbent Assay Kits: USCN Life Science Inc., PRC (Cloud - Clone Corp. USA)’’ has been corrected to: ,,UPIIIa and UPII concentrations were measured in plasma and urine by Enzyme-Linked Immunosorbent Assay (ELISA) using USCN Life Science Inc., PRC (Cloud - Clone Corp. USA) kits.’’

#7: Table 1; A lower part of Table 1 seems to a duplication of upper part of Table 1.

Response 7: Table 1, which was duplicated by mistake, has been corrected.

#8: Urolithiasis contains renal stone, ureter stone, bladder stone, and urethral stone. Did concentrations of UPs change according to the position of urolith?

Response 8: In our study, we did not notice any significant differences in the concentration of UPs in patients with urolithiasis depending on the location of the stones. Due to the small number of patients with urolithiasis, we want to continue our research. As we mentioned in the title, these are preliminary studies to assess UPs in benign urological diseases.

#9: Table 2: At the first column, BHP should be revised as BPH.

Response 9: Of course, the shortcut has been corrected.

#10: Table2 and Table 3: P values should be provided in comparison to control versus each pathological condition.

Response 10: P values were calculated using one-way ANOVA analysis of variance. The one-way analysis of variance (ANOVA) is used to determine whether there are any statistically significant differences between the means of three or more independent (unrelated) groups. In our study, we checked the differences between mean UPs in 3 independent groups: BPH, US and C1 (only men) and UTI, urolithiasis and C2 (men and women), therefore a single p value is presented in the table. Further post-hoc analysis (Tukey's post-hoc test) makes it possible to determine p in comparison to control versus each pathological condition as described in the text below the tables. To better visualize the results (p values) obtained on the post-hoc basis, Tables 2 and 3 have been extended.

#11: Line261-263; In relation to the comment of #10, did this sentence correctly explain the contents of Table 3? Except for plasma UPII, all the p values were less than 0.05.

Respons 11: We hope the explanation given above, to item 10 has also clarified the Reviewer's doubts regarding item 11.

#12: Figure 1: I think that Figure 1 repeats the explanation of Table 2 and Table 3.

Responsee 12: Of course, Figure1 is a visualization of the values shown in Tables 2 and 3, so we removed them.

#13: After treatment of benign diseases of the urinary tract, will UPs decrease to normal range?

Response 13: This is a very good question. As we are only at the initial stage of research on UPs in urological diseases and we have not yet collected an appropriate group of cured patients, at the moment we cannot answer to this  reviewer's interesting reviewer's question. We hope to describe such results in the next article.

#14: How did you estimate sample size to investigate the utility of UPs in the condition of benign diseases of the urinary tract?

Response 14: For the determination of UPIIIa and UPII, we used both urine and plasma samples in a volume of 100 µL for each test. To determine urine creatinine, into which urinary UPs levels were converted, we needed 250uL of a urine sample.

Round 2

Reviewer 1 Report

Thanks

No comments and suggestions for the authors

Author Response

Response:

We would like to thank the Reviewer for the positive assessment of our manuscript and for accepting our explanations.

Reviewer 2 Report

Thanks for shortening your introduction. This made your paper more clear cut. Please add a statement of the acutal and potential clinical relevance of your findings in discussion and conclusion. 

Author Response

Response:

We would like to thank the Reviewer for accepting the amendments to our manuscript. As suggested by the reviewer, we have added information on the clinical significance of UPs in urological diseases.

We added in discussion: page 13. Lines 505-508 (all adjustments)

[The obtained preliminary results of UPs in the urine and plasma of patients with selected urological diseases show that these proteins may be helpful as potential biomarkers for monitoring the course of urological diseases and assessing the effectiveness of treatment of these diseases.]

We added inconclusion: page 13. Lines 515-517 (all adjustments)

[UPs could play a role as potential biomarkers for assessing the course of urological diseases and the effectiveness of their treatment.]

Reviewer 3 Report

The authors well addressed the issues I had pointed out.

Author Response

Response:

We would like to thank the Reviewer for accepting the amendments to our manuscript.